# Iron Deficiency and Anemia in Male and Female Adolescent Athletes Who Engage in Ball Games

**DOI:** 10.3390/jcm12030970

**Published:** 2023-01-27

**Authors:** Daniela Nicotra, Rakefet Arieli, Noam Redlich, Dalya Navot-Mintzer, Naama W. Constantini

**Affiliations:** 1Faculty of Medicine, The Hebrew University, Jerusalem 9112102, Israel; 2Shaare Zedek Medical Center, Hebrew University, Jerusalem 9103102, Israel; 3The Ribstein Center for Sports Medicine and Research, Wingate Institute, Netanya 42902, Israel

**Keywords:** iron deficiency, ferritin, anemia, athletes, adolescents

## Abstract

The aim of this study was to assess the prevalence of iron deficiency (ID) and iron deficiency anemia (IDA) among male adolescent athletes who participate in non-calorie-restricting sports, and to compare the results with female athletes of the same age and sports. Data of the hemoglobin concentration (Hb) and serum ferritin (sFer) levels of male (*n* = 350) and female (*n* = 126) basketball and football players, aged 11–18, from two sport medicine centers in Israel were gathered and analyzed. Mild ID was defined as sFer ≤ 30 µg/L, moderate as sFer ≤ 20 µg/L, and severe as sFer ≤ 10 µg/L. IDA was defined as sFer ≤ 20 µg/L and Hb < 13 g/dL for males and sFer ≤ 20 µg/L and Hb < 12 g/dL for females. The prevalence of mild ID was 41.1% and 53.2%, moderate was 17.4% and 27.8%, and severe was 2% and 4.8% in males and females, respectively. The prevalence of IDA was 2.6% in males and 4% in females. Mild and moderate ID was significantly higher among females. In conclusion, non-anemic ID, which is known to be common among female athletes, especially in sports requiring leanness, is also highly prevalent among adolescent males playing ball games. Therefore, screening for hemoglobin and sFer is recommended for young athletes of both genders and in all sports.

## 1. Introduction

Iron deficiency (ID) and iron deficiency anemia (IDA) are the most common micronutrient disorders in the world [1]. Iron is a crucial component of cell function. It is a part of the hemoglobin molecule within the erythrocytes, which allows oxygen transportation and utilization. Iron is also found in different compounds in the body, such as myoglobin, various cytochrome enzymes, and mitochondria. With the decrease in iron stores, the cells lose their capacity for electron transport and energy metabolism. In the erythrocytes, hemoglobin synthesis is impaired, which eventually causes IDA [2,3].

Serum ferritin (sFer) levels reflect total body iron stores. The cutoff point varies for the sFer value under which iron stores are defined as deficient. In the majority of studies, the cutoff point is below 30 µg/L [2,4,5,6,7]. However, there are other cutoff points, such as 35 [8,9], 25 [10], 20 [11], and 16 µg/L [12]. In some studies, a definition of mild, moderate, and severe ID is given (sFer below 30, 20, and 10 µg/L, respectively) [2,13,14]. The WHO defines anemia as Hb < 13 g/dL for males and Hb < 12 g/dL for females. IDA is defined as sFer < 15 µg/L and Hb < 13 g/dL or Hb < 12 g/dL for males and females, respectively [1]. However, in many studies, especially those dealing with athletes, IDA is defined as sFer < 20 µg/L and Hb < 13 g/dL or Hb < 12 g/dL for males and females, respectively [15,16,17,18].

The clinical features of ID in the general population are reduced physical capacity as well as decreased cognitive functioning [1,19]. In the athlete population, various studies have shown that ID with or without anemia can negatively affect aerobic capacity and exercise performance [4,8,11,13,20,21,22].

In females, there is a higher prevalence of ID and IDA compared with males, mainly due to iron loss in menstruation [1,13,23]. However, poor nutrition—with or without eating disorders—such as often seen in esthetic sports, can lead to nutritional deficiencies, including ID [24]. In endurance athletes of both genders, a calorie-restricted diet also causes a higher prevalence of ID, mainly because of high training stress and insufficient intake of food and nutrients [5,10].

In addition to insufficient intake, high-intensity sport can lead to ID through other mechanisms, such as increased iron demand, higher iron loss due to micro-trauma and ischemia of the gut, hemolysis, iron loss through increased sweating [12,18,25], and dysregulation of iron absorption and storage by elevated hepcidin levels post-exercise [26].

The prevalence of ID and IDA in adolescent athletes and non-athletes is reported in the literature with a wide range, such as gender, type of sport, and ID definition. The reported incidence of ID among menstruating adolescent females in the general population is 15–36% [2,27], whereas in the same age group of female athletes, the reported incidence is 52–86% [6,28]. Young males have been studied less frequently for ID, with findings of a prevalence of 4–15% among adolescent males [29,30] and 18–65% among well-trained newly recruited soldiers and athletes [6,31]. The prevalence of IDA is approximately 10% for females and 4% for males in industrialized countries [1]. It is important to mention that the prevalence of both ID and IDA is higher in non-industrialized countries [1].

Many studies have been conducted regarding athletes’ ID and IDA, focusing mainly on females and endurance athletes. The prevalence of ID and IDA varies between studies due to the study subjects’ differences in terms of age, type of sport, and performance level, and also due to the different definitions of ID. In some studies, ID is defined as a ferritin level below 12, whereas in others, the level can be 16, 20, 25, 30, or 50 [5,10,12,13,17,20,21,32,33,34].

There is very little information regarding the prevalence of ID among adolescent male athletes engaged in non-calorie-restricting sports, such as ball games [35]. Therefore, in the current study, we examined the prevalence of ID, anemia, and IDA in male adolescent basketball (BB) and football (FB) players and compared the data with the prevalence in female ball game players of the same age.

## 2. Methods Section

### 2.1. Study Design

A cross-sectional study was performed using data (blood count and ferritin level) collected from medical files of two sport medicine clinics in Israel between 2017 and 2020: The Heidi Rothberg Sport Medicine Center at Shaare Zedek Medical Center in Jerusalem and The Ribstein Center for Sports Medicine and Research at the Wingate Institute.

#### 2.1.1. Participants

The study population included 350 adolescent male athletes between the ages of 11–18 (mean age 14.5 ± 1.7): 164 football (FB) players and 186 basketball (BB) players; and 126 female athletes of the same ages (mean age 14.8 ± 1.2): 29 FB players and 97 BB players. All participants trained at least five times a week for at least 90 min per session and were playing in a competitive league. Athletes for whom data regarding iron store levels as reflected in their serum ferritin (sFer) levels were not available were excluded from the study.

#### 2.1.2. Primary Outcome Measures

The evaluation of ID, anemia, and IDA was based on measurements of complete blood count and iron stores (hemoglobin concentration and sFer levels). In addition, other values of blood count, such as hematocrit (HCT), mean corpuscular hemoglobin (MCH), mean corpuscular volume (MCV), and red blood cells (RBC), are depicted in the study. The blood samples were collected from the athletes over the course of the specified years. Venous blood was drawn from the athletes in one of four HMOs that exist in Israel and analyzed in the corresponding laboratory. The results were recorded in g/dL for hemoglobin and ng/mL for sFer. The data were collected from the athletes’ medical files anonymously, ensuring the confidentiality of the athletes. Iron deficiency was defined as sFer ≤ 30 µg/L based on prior studies [2,5,6,8] and was further divided into three groups: mild ID–below 30 µg/L; moderate ID–below 20 µg/L; and severe ID–below 10 µg/L [2,6,14]. Anemia was defined as Hb < 13 g/dL for males and Hb < 12 g/dL for females. IDA was defined as sFer ≤ 20 µg/L and Hb < 13 g/dL or Hb < 12 g/dL for males and females, respectively [5,8,32]. The variables were compared between male and female athletes. Due to the wide range of athletes’ ages, the variables were also analyzed by dividing the data into two age groups: 11–14 and 15–18 years.

### 2.2. Statistical Methods

Statistical analysis was performed using SPSS (SPSS v.27.0; IBM Corp., Armonk, NY, USA). Based on the empirical data, the percentage of ID, anemia, and IDA were estimated, and a 95% confidence interval was calculated. In order to compare the categorical variables between the male and female athletes, and to make a sub-analysis according to age groups, the χ^2^ and the Fisher’s exact test were used. The comparison of quantitative variables between the two groups of athletes (male and female) was carried out using the two-sample *t* test.

### 2.3. Ethical Approval

The study was carried out in accordance with the standards of ethics outlined in the Declaration of Helsinki. The study was approved by the Helsinki Ethics Committee in Shaare Zedek Medical Center, Israel (approval number: 0074-20-SZMC).

## 3. Results

### 3.1. Blood Count and Iron Status among Adolescent Athletes

Gender, age, red blood cell parameters, and iron status are presented in Table 1. The prevalence of ID among male adolescent athletes playing FB and BB was divided into mild, moderate, and severe ID. The prevalence of mild ID (sFer ≤ 30 µg/L) was 41.1% (*n* = 144), moderate ID (sFer ≤ 20 µg/L) was 17.4% (*n* = 61), and severe ID (sFer ≤ 10 µg/L) was 2% (*n* = 7). IDA (sFer ≤ 20 µg/L and Hb < 13 g/dL) was 2.6% (*n* = 9).

Among the female adolescent athletes’ group, the prevalence of mild ID (sFer ≤ 30 µg/L) was 53.2% (*n* = 67), moderate ID (sFer ≤ 20 µg/L) was 27.8% (*n* = 35), and severe ID (sFer ≤ 10 µg/L) was 4.8% (*n* = 6). IDA (sFer ≤ 20 µg/L and Hb < 12 g/dL) prevalence was 4% (*n* = 5).

No difference was found in the prevalence of ID and IDA between BB players and FB players in either gender.

### 3.2. Comparison between Males and Females

Differences were found in the prevalence of ID and IDA between male and female adolescent athletes. The prevalence of mild and moderate ID was significantly higher in the female group. Severe ID and IDA prevalence was also higher among females but were not statistically significant (Figure 1).

### 3.3. Sub-Analysis according to Age Groups

The athletes in the study were between the ages of 11–18 years. A sub-analysis was performed to examine the data according to age groups. Mild, moderate, and severe ID, as well as ID anemia, were analyzed for two age groups: ages 11–14 years and ages 15–18 years, as depicted in Table 2.

In the younger age group, there were no statistically significant differences in the prevalence of ID between male and female adolescent athletes. However, in the older age group, there were statistically significant differences in the prevalence of mild and moderate ID in female athletes. There was no statistical significance in the prevalence of ID anemia between genders in both age groups.

## 4. Discussion

This cross-sectional study demonstrates a high prevalence of ID among male adolescent athletes who participate in ball games—sport disciplines that do not require lean body mass and/or caloric restriction.

A vast number of studies have examined iron status in endurance sports, such as running and triathlon, but the participants were all above 18. In these studies, the prevalence of ID in male and female athletes ranged from 30–60% [8,10].

We did not find any study that looked into the prevalence of ID in adolescent male athletes who engage in non-calorie-restricting sports. There have been very few studies on young male athletes that include diverse sport disciplines and levels or various ages, including adults. The prevalence of ID ranged from 19 to 65% in these studies, as depicted in Table 3.

In this study, we tested 350 male adolescent athletes playing FB and BB at a competitive level. The results showed a very high prevalence of ID (41%) among those athletes. This finding is new and important, since there are no recommendations for the routine testing of iron level status in adolescent male athletes, including for those playing ball games.

ID affects physical, cognitive, and mental performance [8,22]; therefore, unawareness and undertreatment of ID can be a major factor influencing success.

In the second part of the study, we compared male and female adolescent athletes. There is a high number of studies concerning female adolescent athletes and ID [12,16,17,28,33]; moreover, the high level of interest in the iron status of female athletes is not in vain, as there is a high prevalence of ID and IDA in this group. This is due to the greater iron loss among menstruating females, and a lower calorie intake, which is more prevalent among female athletes [13,21]. Many studies also showed the benefits of iron supplementation for iron-depleted female athletes in order to improve physical performance [8,11,21,34].

This study shows again that female adolescent athletes have a high prevalence of mild and moderate ID.

A sub-analysis was conducted on two age groups: 11–14 years and 15–18 years. The analysis revealed statistically significant differences in the prevalence of mild and moderate ID in female athletes in the older age group. It is likely that this difference is related to the increased frequency and intensity of menstruation during the years of adolescence. However, there were no statistically significant differences in the prevalence of mild, moderate, and severe ID in the younger age group and in ID anemia in both age groups between genders. This finding is noteworthy as it demonstrates that adolescent male athletes of any age are at risk of suffering from ID or ID anemia throughout adolescence. Adolescent males, unlike young children, adolescent females, or pregnant women, are not considered to be at risk for ID [29,37,38]. We assume that the increased physical demands caused by body growth in male adolescent athletes—including muscles, bones, and plasma volume increases [6,9]—together with imbalanced eating habits and the consumption of foods that have low iron concentration, can cause ID [1,3]. This cross-sectional study did not focus on the various dietary habits and preferences that may be a major contributing factor to ID in the study population. In Israel, similar to the rest of the world, the number of vegetarian and vegan people is rising—due to ethical, ecological, and health reasons [27,30,39]. This trend has increased the risk of ID, as meat products are rich in iron.

In addition to the higher iron requirement during adolescence and the potentially insufficient dietary intake, there are other mechanisms by which exercise can lead to ID–such as increased iron loss due to ischemia of the gut, microtrauma, hemolysis, and loss through increased sweating [12,18,25].

In recent years, it has become apparent that post-exercise elevated hepcidin levels reduce iron absorption; therefore, the timing of iron supplementation plays an important role [14,40].

In the literature, there is no consensus regarding a cutoff point of sFer levels for the definition of ID, which makes it difficult to compare between studies. Numbers vary from 10–50 µg/L [5,10,12,13,17,20,21,33,34], and some studies suggest different numbers according to age and gender [5]. We chose, similar to most studies, a cutoff of 30 µg/L—a level that has been shown to reduce aerobic performance and work capacity [8].

The prevalence of anemia (Hb < 13 g/dL) among male adolescent athletes in this study was 10.4%, a higher prevalence than the reported value in the normal worldwide population—4% [1]. The prevalence of anemia was similar between female and male athletes. In the current study, we focused mainly on iron stores, but there may be other deficiencies (folic acid, B12 deficiency) or hemoglobinopathies (such as thalassemia traits and sickle cell anemia) that could explain anemia occurring without iron deficiency.

In summary, in the current study, which investigated iron status among hundreds of adolescent male athletes involved in sports that do not require caloric restriction, a very high prevalence (41%) of ID was found. Due to the impact of ID even without anemia on health and performance, we recommend an annual check-up of blood count and iron stores in adolescent athletes of both genders. In cases of deficiency, the etiology should be determined and treated accordingly.

## 5. Study Limitations

The study was cross-sectional, and there was no control group of non-athletes. Furthermore, 60% of the female athletes and 5% of the males who trained at the Wingate Institute received nutritional guidance over the years, including iron supplementation, which could have affected the results, especially for females. As this was a cross-sectional study with data collected from medical files, it was not possible to include data about eating habits from questionnaires. The inclusion of this information would have greatly contributed to the research and provided a deeper understanding of the reasons behind the high prevalence of ID in the study’s population. Lastly, the blood analysis was performed in several labs according to the athlete’s HMO.

## 6. Future Research

Future research is needed for a better understanding of the etiology of iron deficiency in healthy, adolescent male athletes. Is ID a consequence of low dietary iron intake, increased physiological demands, iron loss during exercise, or disturbed absorption due to increased hepcidin levels post-physical effort? Guidelines for the prevention, detection, and treatment of iron deficiency in adolescent male athletes should also be examined.

## Figures and Tables

**Figure 1 jcm-12-00970-f001:**
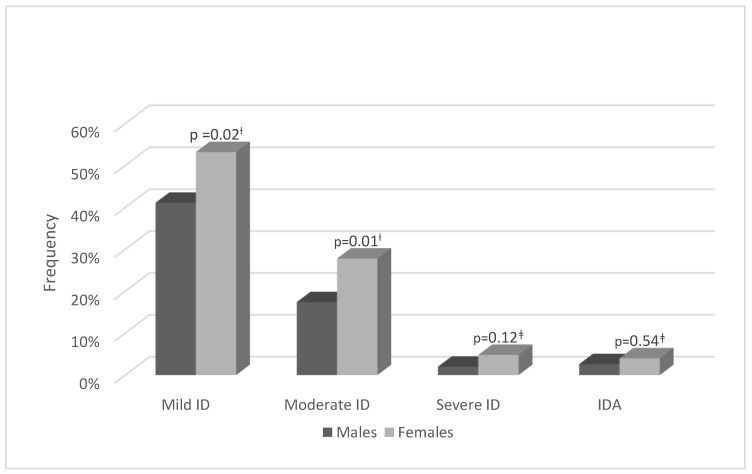
Males’ and females’ iron deficiency and iron deficiency anemia frequencies and a comparison between gender groups. Mild ID = sFer ≤ 30 µg/L; moderate ID = sFer ≤ 20 µg/L; severe ID = sFer ≤ 10 µg/L. IDA = sFer ≤ 20 µg/L and Hb < 13 g/dL or Hb < 12 g/dL for males and females, respectively. ID and IDA were calculated with a 95% confidence interval. † Indicates the use of χ^2^ test. ‡ Indicates the use of Fisher’s exact test.

**Table 1 jcm-12-00970-t001:** Gender, age, red blood cell parameters, and iron status data.

Gender		Age (Years)	HB (g/dL)	HCT (%)	RBC (M/μL)	MCV (fL)	MCH (pg)	Serum Ferritin (ng/mL)
Male	*n*	350	347	346	347	347	347	350
	Mean	14.6	14.2	42.7	5.0	85.3	28.4	43.2
	Median	15.0	14.2	42.7	5.0	85.1	28.4	35.2
	Standard Deviation	1.7	1.0	2.9	0.3	4.3	1.6	30.8
	Minimum	11.0	9.3	29.7	4.0	68.0	21.0	8.0
	Maximum	18.0	17.7	50.8	6.0	97.0	32.8	260.0
Female	*n*	126	125	77	90	77	76	126
	Mean	14.8	13.0	39.5	4.6	86.1	28.3	35.1
	Median	15.0	13.1	39.4	4.6	87.3	28.5	30.0
	Standard Deviation	1.3	1.0	2.7	0.3	5.8	2.2	26.4
	Minimum	12.0	9.7	31.6	3.8	61.0	18.8	4.9
	Maximum	18.0	15.7	45.7	5.5	96.0	31.8	165.0

HB = hemoglobin; HCT = hematocrit; MCH = mean corpuscular hemoglobin; MCV = mean corpuscular volume; RBC = red blood cells.

**Table 2 jcm-12-00970-t002:** Prevalence of mild, moderate, severe ID and ID anemia in male and female adolescent athletes according to age groups.

	Mild ID (%)			Moderate ID (%)			Severe ID (%)			ID Anemia (%)		
	M	FM	*p*-Value	M	FM	*p*-Value	M	FM	*p*-Value	M	FM	*p*-Value
Age ≤ 14 years	51.8	60.0	0.29 †	23.5	29.1	0.4 †	3.6	5.5	0.69 ‡	3.6	5.5	0.69 ‡
Age ≥ 15 years	31.5	47.9	0.01 †	12.0	26.8	0.01 †	0.5	4.2	0.06 ‡	1.7	2.9	0.62 ‡

ID = iron deficiency; M = male; FM = female. Mild ID = sFer ≤ 30 µg/L; moderate ID = sFer ≤ 20 µg/L; severe ID = sFer ≤ 10 µg/L. IDA = sFer ≤ 20 µg/L and Hb < 13 g/dL or Hb < 12 g/dL for males and females, respectively. † Indicates the use of χ^2^ test. ‡ Indicates the use of Fisher’s exact test.

**Table 3 jcm-12-00970-t003:** Iron deficiency in male adolescent athletes.

Author	*n*/Study Population	Participants’ Ages	Sport Types	Definition of ID	Prevalence of ID
Toivo et al., 2020 [7]	261	14–17	Various sports (11 types)	sFer < 30 µg/L	27%
Shoemaker et al., 2019 [6]	69	8–16	School- or club-sponsored sport	sFer < 30 µg/L	65%
Constantini et al., 2000 [35]	43	12–18	Gymnasts, swimmers, and tennis and table-tennis players	sFer < 20 µg/L	36% in gymnasts,20% in the rest
Dubnov and Constantini, 2004 [36]	32	14–18	Basketball	sFer < 20 µg/L	19%

ID = iron deficiency; sFer = serum ferritin.

## Data Availability

The data presented in this study are available on request from the corresponding author.

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
