# Peer review of "Iron Deficiency and Anemia in Male and Female Adolescent Athletes Who Engage in Ball Games"

_jcm, 2023, doi:10.3390/jcm12030970_

Round 1

Reviewer 1 Report

The present study investigates in a cross-sectional design, the prevalence of iron deficiency and anaemia in adolescent athletes engaged in ball games. This is an important under-investigated topic as the available results indicate that anaemia is a problem in these age groups. Hence this is a timely study. It seems like the data-collection is well performed and the ms is well written. A have mostly minor comments and suggestion that might be consider. However, the data collection and analysis procedures must be described in more details. I also suggest exploring the effect of age since the subjects are from a wide age-group (11 – 18 yrs). Also, the title should be modified to include both male and female, as well as anaemia.

Minor comments

Line 17: “….. hemoglobin concentration (Hb) ….”

Line 46: Delete one of the “sFer˂20 µg/L”

Line 65: What do you mean with “it can reach to 52-86%? Do you mean it is reported.

Line 93 - 94: Clarify this sentence (“lacked iron store levels”)

Line 98: “ …. concentration and sFer levels in blood which were drawn from the athletes ….”

Line 97 – 99: Please specify the procedure for blood collection and the methods used for assessment of Hb and sFer. Also include all measured and reported variables.

Line 103: Consider simplify sentence: “IDA was defined as anemia and sFer≤20 µg/L [5,8,32]”

Line 104 – 105: consider deleting sentence.

Line 138: Consider rewrite: “…. not statistically significant (p values in figure 1).”

Line 139-140: Delete sentence. 

Figure 1: The figure needs a new printable layout. Please also include SD or CI.

Line 144: Change to “Hb concentration (Hb) and sFer”.

Line 148 – 152: This information can be deleted (already in the body text and figure)

Author Response

Many thanks for taking the time to review our work. Please find the attached file for your reference.

Reviewer 2 Report

The title of the article is not innovative, not something new

I do not think it adds anything to the scientific community

The references' number should be definitely increased

The structure and the language are fine

The main question is the relation between iron deficiency and male athletes in ball games

I have already mentioned in my first evaluation that this specific manuscript does not add anything new in science

There is definitely iron deficiency in athletes

I have observed in other publications that this specific topic has already been analyzed

The topic is relevant to the field but ιτ does not fill in or specifies any gaps in the field

 the references are relevant but they describe almost the same topic

I have no comments to make on the figures-tables

Regarding the conclusion i can actually detect any inovatine results

Author Response

(The authors gave the same response as above.)

Reviewer 3 Report

The manuscript entitled “Iron Deficiency in Male Adolescent Athletes Who Engage in Ball Games” is well written and properly explained. Iron deficiency can result in major health issues and should be considered for routine screening of players. Adolescent players, especially male players of ball games are usually neglected for Iron deficiency but the present study has shown the importance of Iron deficiency in adolescent male athletes. Introduction of the manuscript gives important information about Iron deficiency in male and female players with detailed references. Material and methodology is very well written and easy to understand. Results are given in proper manner and discussed in detail with good references. Shortcomings of the study with proper reasons are discussed, whereas the future suggestions has also been given. Best wishes and regards.

Author Response

(The authors gave the same response as above.)
